# Lipopolysaccharide-Induced Strain-Specific Differences in Neuroinflammation and MHC-I Pathway Regulation in the Brains of Bl6 and 129Sv Mice

**DOI:** 10.3390/cells11061032

**Published:** 2022-03-18

**Authors:** Maria Piirsalu, Keerthana Chithanathan, Mohan Jayaram, Tanel Visnapuu, Kersti Lilleväli, Mihkel Zilmer, Eero Vasar

**Affiliations:** 1Institute of Biomedicine and Translational Medicine, Department of Physiology, University of Tartu, 19 Ravila Street, 50411 Tartu, Estonia; keerthana.chithanathan@ut.ee (K.C.); mohan.jayaram@ut.ee (M.J.); tanel.visnapuu@ut.ee (T.V.); kersti.lillevali@ut.ee (K.L.); eero.vasar@ut.ee (E.V.); 2Center of Excellence for Genomics and Translational Medicine, University of Tartu, 50411 Tartu, Estonia; mihkel.zilmer@ut.ee; 3Institute of Biomedicine and Translational Medicine, Department of Biochemistry, University of Tartu, 19 Ravila Street, 50411 Tartu, Estonia

**Keywords:** lipopolysaccharide, 129S6/SvEvTac, C57BL/6NTac, MHC-I, neuroinflammation, innate immunity

## Abstract

Many studies have demonstrated significant mouse-strain-specific differences in behavior and response to pathogenic and pharmacological agents. This study seeks to characterize possible differences in microglia activation and overall severity of neuroinflammation in two widely used mouse strains, C57BL/6NTac (Bl6) and 129S6/SvEvTac (129Sv), in response to acute lipopolysaccharide (LPS) administration. Locomotor activity within the open field arena revealed similar 24 h motor activity decline in both strains. Both strains also exhibited significant bodyweight loss due to LPS treatment, although it was more severe in the Bl6 strain. Furthermore, LPS induced a hypothermic response in Bl6 mice, which was not seen in 129Sv. We found that 24 h LPS challenge significantly increased the inflammatory status of microglia in 129Sv mice. On the other hand, we observed that, under physiological conditions, microglia of Bl6 seemed to be in a higher immune-alert state. Gene and protein expression analysis revealed that LPS induces a significantly stronger upregulation of MHC-I-pathway-related components in the brain of Bl6 compared to 129Sv mice. The most striking difference was detected in the olfactory bulb, where we observed significant LPS-induced upregulation of MHC-I pathway components in Bl6 mice, whereas no alterations were observed in 129Sv. We observed significant positive correlations between bodyweight decline and expressions of MHC-I components in the olfactory bulbs of Bl6 mice and the frontal cortex of 129Sv, highlighting different brain regions most affected by LPS in these strains. Our findings suggest that the brains of Bl6 mice exist in a more immunocompetent state compared to 129Sv mice.

## 1. Introduction

Since mice share many physiological and genetic traits with humans, they have become extensively used model organisms for modeling human diseases. However, difficulties in translating the data from animal studies to human physiology and pathogenesis questions the use of a single strain to address several issues. Inbred mouse strains are essential animal models in scientific laboratories in many research fields. These mice are homozygous at all genetic loci, and the expression of recessive alleles results in the formation of a unique phenotype for every strain. Thus, one must keep in mind the characteristic differences among strains when selecting an appropriate model. Genetic differences among strains often lead to differential responses in models of human disease, and it is therefore extremely important to obtain new knowledge of strain-specific differences to enrich our understanding of appropriate model selection, which is the key to effective data interpretation.

C57BL/6NTac (Bl6) and 129S6/SvEvTac (129Sv) belong to the most widely used mouse strains in biomedical and transgenic research and are the gold standard when generating transgenic mouse models. These two genetically distinct inbred mouse strains differ from one another significantly in several ways. It is well-established that Bl6 mice are more active and have increased exploratory behavior, while 129Sv mice are less active and display higher levels of anxiety [1,2,3]. Previous studies have demonstrated Bl6 mice to have an active coping strategy in stressful situations, whereas the coping strategy of 129Sv strain is more passive [1,3,4].

Bl6 and 129Sv are also widely used in immunological studies. Several studies have demonstrated that they display dissimilarities in immune responses and have differential susceptibility to infectious diseases. For example, influenza virus infection causes a hyper-induction of proinflammatory cytokines in 129Sv, and mice tend to die from infection. However, the proinflammatory cytokine response in the Bl6 strain is modest, and mice are able to cope effectively with the infection [5]. In addition, we have recently shown that systemic administration of the inflammation-inducing major compound of Gram-negative bacteria cell walls, LPS, leads to hypometabolism in the Bl6 strain but enhances the production of proinflammatory metabolites in 129Sv mice [6]. Furthermore, we demonstrated that LPS administration leads to changes in polyamine metabolism in Bl6 mice but not in 129Sv mice. More precisely, LPS increased the biosynthesis of putrescine in Bl6 mice, which has been shown to possess neuroprotective activity in the central nervous system (CNS) [7]. This led us to investigate further the possible different outcomes of LPS-induced inflammatory response in the CNS of Bl6 and 129Sv mice.

Despite the reported variations in immune response between Bl6 and 129Sv, most of the studies have been focused mainly on the MHC-II pathway, and, to our knowledge, no previous study has analyzed strain differences of the MHC-I pathway in response to LPS. Thus, to better understand the underlying genetic differences in immunity and defense, we studied these two well-characterized inbred mouse strains utilizing a series of approaches—including open field test, bodyweight and temperature measurements, flow cytometry, RT-qPCR, and Western blotting—to examine the profile of microglia activation and overall severity of neuroinflammatory status.

## 2. Materials and Methods

### 2.1. Animals and Their Distribution into Experimental Groups

Wild-type male mice (16–23-weeks-old) from the two inbred strains, C57BL/6NTac (Bl6, *n* = 58) and 129S6/SvEvTac (129Sv; *n* = 58), were bred and housed in the Laboratory Animal Center at University of Tartu. Mice from both strains were randomly assigned to different experimental groups (Figure 1).

The first cohort (Bl6, *n* = 16 129Sv, *n* = 16) was used to measure locomotor activity after LPS or saline administration during the 24 h period.

The second cohort (Bl6, *n* = 20; 129Sv, *n* = 20) of mice was used to characterize LPS-induced changes in the brain by qPCR for various inflammatory markers. Animals were sacrificed by decapitation, and various brain regions were dissected for gene expression analysis.

The third cohort (Bl6, *n* = 10; 129Sv, *n* = 10) of mice was used to determine LPS-induced strain-specific differences in microglial profile by flow cytometry. Mice were euthanized by carbon dioxide (CO_2_) overdose, and the hippocampus and cerebellum were dissected for further flow cytometric analysis.

The fourth cohort (Bl6, *n* = 12 129Sv, *n* = 12) of mice was used for Western blot analysis to confirm our findings from the gene expression analysis. Animals were sacrificed by decapitation, and various brain regions were dissected for protein expression analysis.

Mice were kept under standard conditions with unlimited access to food and water on a 12/12 h light/dark cycle.

### 2.2. LPS Treatment

LPS (derived from *E. coli* serotype 0111:B4; Sigma-Aldrich, St. Louis, MO, USA) was dissolved in 0.9% NaCl (saline). In each cohort, one group was injected intraperitoneally (i.p.) with 0.5 mg/kg bodyweight of LPS solution and another group received injection of an equal volume of 0.9% NaCl (saline) as controls.

### 2.3. Bodyweight and Temperature Determination

Baseline body temperature readings were obtained before LPS or saline administration using a rectal thermometer (TSE Technical and Scientific Equipment GmbH, Berlin, Germany). After 24 h of LPS and saline administration, the body temperature measurement was repeated. Body temperature data was obtained from the first and second cohort of mice. Bodyweight was measured before and 24 h post LPS and saline injection in all cohorts.

### 2.4. Locomotor Activity

The first cohort of mice was used for locomotor activity assessment. The effect of LPS on locomotor activity was monitored for 24 h in PhenoTyper^®^ (EthoVision 3.0, Noldus Information Technology, Wageningen, The Netherlands) cages. Animals were individually housed in 30 cm × 30 cm × 35 cm plexiglass cages with sawdust bedding. Mice had free access to food and water throughout the testing period. Mice were kept under a 12:12 h light/dark cycle. Each cage was equipped with a top unit with integrated infrared sensitive camera and infrared LED lights, which makes tracking possible during the dark phase. The open-field arena was virtually divided into central and peripheral zones. The center zone was defined as half of the overall area of the test arena. Total distance traveled and time spent moving in the whole arena, central, and peripheral zones were measured. Animal movements were continuously recorded by a video-tracking system.

### 2.5. Total RNA Isolation cDNA Synthesis and Real-Time qPCR

The second cohort of mice was used for qPCR analysis. Total RNA from olfactory bulbs, prefrontal cortex, hypothalamus, hippocampus, midbrain, and cerebellum was extracted by using Trizol^®^ Reagent (Invitrogen, Waltham, MA, USA) according to the manufacturer’s protocol. RNA was reverse transcribed (2 µg) with random hexamers (Applied Biosystems, Foster City, CA, USA) and SuperScript™ III Reverse Transcriptase (Invitrogen, USA). Real-time qPCR was performed by using 5x HOT FIREPol^®^ EvaGreen^®^ qPCR Supermix (Solis BioDyne, Tartu, Estonia) and corresponding primers (TAG Copenhagen, Copenhagen, Denmark; Table 1) on a QuantStudio 12 K Flex Real-Time PCR Detection System (Applied Biosystems, USA) according to the manufacturer’s instructions.

### 2.6. Flow Cytometry

The third cohort of mice was used for flow cytometry analysis. After 24 h of LPS/saline challenge, mice were euthanized by carbon dioxide (CO_2_) overdose, and the hippocampus and cerebellum were dissected. Dissected brain regions were gently homogenized through 70 µm cell strainers (BD Falcon) in FC buffer (ice-cold, phosphate-buffered saline (PBS) + 1% fetal calf serum). Isolated cells were blocked with 10% rat serum in ice-cold PBS for 1 h. Brain cells were stained with 0.5 μL of anti-mouse CD11b-BV421 (#101251, BioLegend, San Diego, CA, USA), CD45-BV650 (#103151, BioLegend, San Diego, CA, USA), O4-PE (#130-117-357, Miltenyi Biotec, Bergisch Gladbach, Germany), CX3CR1-A488 (#149021, BioLegend San Diego, CA, USA) for 1 h at 4 °C. Cells were washed and resuspended into 0.5 mL PBS. Samples were acquired with a 5-laser LSR Fortessa (BD Biosciences, San Jose, CA, USA) cytometer and analyzed with Kaluza v1.2 software (Beckman Coulter, Indianapolis, IN, USA).

### 2.7. Protein Extraction and Western Blotting

Fourth cohort of mice was used for Western blot analysis. Total protein fraction was extracted from hippocampus and olfactory bulbs. Samples were sonicated in ice-cold RIPA lysis buffer (Thermo Scientific, Waltham, MA, USA) containing 1× protease inhibitor (78430, Thermo Scientific) and centrifuged at 14,000× *g* for 10 min at 4 °C. Protein concentrations were determined with the BCA protein assay kit (23225, Thermo Scientific). Equal amounts of protein (20 μg) from each sample were separated on a NuPAGE Bis–Tris gel using the XCELL SureLock System (Invitrogen, USA). Thereafter, protein was transferred to a nitrocellulose membrane, after which the membranes were blocked and incubated with primary antibodies (Table 2) overnight at 4 °C. Immunoblots were then incubated with goat anti-rabbit (A11369, Invitrogen, USA) or goat anti-mouse (A-21057, Invitrogen, USA) fluorescent conjugated secondary antibodies for 1 h at room temperature, followed by visualization using a LI-COR Odyssey CLx system (LI-COR Biotechnologies, Lincoln, NE, USA). B-actin was used as a loading control.

### 2.8. Statistical Analysis

Results are expressed as mean values ± SEM. Statistical analyses were performed using GraphPad Prism 9 software (GraphPad, San Diego, CA, USA). The Shapiro–Wilk test was applied to assess the normality of data distribution. Locomotor data was log10-transformed prior to statistical analysis to make the data correspond to a normal distribution. Bodyweight (ΔBW) and temperature (ΔBT) loss were expressed as percent change of the initial weight and temperature, respectively (ΔBW = (final weight − initial weight)/initial weight × 100%; ΔBT = (final temperature − initial temperature)/ initial temperature × 100%). Statistical analysis was performed using two-way ANOVA (strain × treatment), followed by Bonferroni post hoc test. All differences were considered statistically significant at *p* ≤ 0.05. The associations between ΔBW, ΔBT, and gene expressions were analyzed using Pearson’s correlation.

### 2.9. Ethics

All animal procedures were performed in accordance with the European Communities Directive (2010/63/EU) and permit (No. 141, 17 April 2019) from the Estonian National Board of Animal Experiments.

## 3. Results

### 3.1. LPS-Implicated Body Temperature and Weight Changes

A single intraperitoneal injection of LPS (0.5 mg/kg) was administered to Bl6 and 129Sv mice, and bodyweight and temperature readings were obtained before and 24 h after LPS and saline administration. Groups were compared using two-way ANOVA (strain × treatment) followed by Bonferroni post hoc test.

Two-way ANOVA demonstrated significant differences in 24 h bodyweight between the LPS- and saline-treated mice in both strains. LPS administration induced a significant bodyweight decrease in both strains (treatment: *F*_(1, 125)_ = 651.9, *p* < 0.0001; strain: *F*_(1, 125)_ = 52.7, *p* < 0.0001; strain × treatment: *F*_(1, 125)_ = 48.19, *p* < 0.0001). However, LPS-induced bodyweight loss was substantially more prominent in Bl6, as the mean bodyweight loss percentage in Bl6 mice was 12.6 ± 0.4%, whereas in 129Sv the decline was merely 7.5 ± 0.3% of the initial bodyweight (Figure 2A).

Strain-dependent effects on the body temperature were observed among LPS-treated animals (treatment: *F*_(1, 68)_ = 12.98, *p* = 0.0006; strain *F*(_1, 68)_ = 5.56, *p* = 0.02; strain × treatment: *F*_(1, 125)_ = 2.3, *p* = 0.13). After 24 h of LPS administration, the body temperature of Bl6 mice dropped significantly (−2.9 ± 0.6%), whereas LPS did not cause a statistically significant change in the body temperature of 129Sv mice (−0.5 ± 0.5%). After administration of LPS, the differences in body temperature between the two strains became statistically significant, indicating a more significant effect of LPS on the Bl6 strain (Figure 2B). A moderate positive correlation (*r* = 0.51, *p* = 0.03) was seen between bodyweight loss and temperature change at 24 h after LPS injection in Bl6 mice, whereas no significant correlation was observed in the 129Sv strain (Figure 2C,D).

### 3.2. LPS-Induced Depression of Locomotor Activity

We next determined LPS-induced alterations in the locomotor activity to establish whether the changes seen in body temperature and bodyweight are related to changes in motor activity. A single intraperitoneal injection of LPS (0.5 mg/kg) was administered to Bl6 and 129Sv mice, and locomotor activity was recorded for 24 h in PhenoTyper cages (Figure 3A). Total distance traveled in the whole arena and in the central and peripheral zones of the arena as well as the time spent moving were recorded. Locomotor data was log10-transformed prior to two-way ANOVA (strain × treatment) analysis to make the data correspond to a normal distribution. Thereafter, Pearson’s correlation analysis was conducted to analyze correlations between 24 h locomotor activity parameters and bodyweight and temperature change.

During the 24 h cycle, LPS induced significant suppression of locomotor activity in the whole arena (treatment: *F*_(1, 23)_ = 38.41, *p* < 0.0001; Figure 3B) as well as in the central zone of the arena (treatment: *F*_(1, 23)_ = 28.58, *p* < 0.0001; Figure 3E) in both mouse strains. When dividing the 24 h cycle into light/dark periods, LPS-induced suppression of locomotor activity in the central zone was only significant during the lights-off period (treatment: *F*_(1, 22)_ = 29.05, *p* < 0.0001; Figure 3F,G), whereas in the whole arena, locomotor activity remained suppressed in light (treatment: *F*_(1, 23)_ = 24.51, *p* < 0.0001; Figure 3C) and dark periods (treatment: *F*_(1, 23)_ = 37.92, *p* < 0.0001; Figure 3D).

As expected, LPS suppressed the time spent moving in both strains, compared to their control counterparts in the whole arena (treatment: *F*_(1, 23)_ = 136.5, *p* < 0.0001; Figure 3H) as well as in peripheral (treatment: *F*_(1, 23)_ = 135.9, *p* < 0.0001; Figure 3I) and central (treatment: *F*_(1, 23)_ = 67.45, *p* < 0.0001; Figure 3J) zones. However, Bl6 control group mice spent significantly more time moving compared to 129Sv control mice in the whole arena, which was even more prominent in the central zone. This probably reflects a higher anxiety level of 129Sv mice.

Pearson’s correlation analysis showed no significant relationship between 24 h locomotor activity parameters and bodyweight and temperature change (Appendix A).

### 3.3. Microglial Profile and Subpopulations

We next questioned whether a systemic immune challenge with LPS may induce a different microglial response in Bl6 and 129Sv strains. The microglial profile was analyzed by flow cytometry of hippocampus and cerebellum tissue 24 h after LPS (0.5 mg/kg; i.p) or saline administration. A combination of cell-specific markers was used to identify CD11b+CD45^intermediate^ microglia, CD11b+CD45^high^ infiltrating myeloid cells (mainly macrophages and neutrophils), and O4+ oligodendrocyte progenitor cells (OPCs). To characterize the activation profile of microglia after LPS stimulation, microglia cells were analyzed for MHC-II and CD206 surface markers. We also investigated CD11b and CX3CR1 expression on microglial cells. Although CD11b is constitutively expressed by microglia, the expression increases upon microglial activation [8,9].

When comparing the control and LPS groups, we found that microglia cell percentages were slightly increased among both strains after 24 h, although this elevation did not reach the statistical significance level in 129Sv mice (cerebellum − treatment: *F*_(1, 15)_ = 16.5, *p* = 0.001; strain: *F*_(1, 15)_ = 8.16, *p* = 0.01; strain × treatment: *F*_(1, 15)_ = 0.6, *p* = 0.44; hippocampus − treatment: *F*_(1, 15)_ = 21.61, *p* = 0.0003; strain: *F*(_1, 15)_ = 0.4, *p* = 0.5; strain × treatment: *F*_(1, 15)_ = 2.49, *p* = 0.14). As expected, the quantities of neutrophils and macrophages were elevated in the cerebellum (treatment: *F*_(1, 15)_ = 51.53, *p* < 0.0001; strain: *F*_(1, 15)_ = 10.10, *p* = 0.006; strain × treatment: *F*_(1, 15)_ = 0.53, *p* = 0.48) and hippocampus (treatment: *F*_(1, 15)_ = 43.43, *p* < 0.0001; strain: *F*_(1, 15)_ = 10.04, *p* = 0.007; strain × treatment: *F*_(1, 15)_ = 1.49, *p* = 0.24) after LPS administration in both strains.

As previous studies have demonstrated that LPS-activated microglia mediate oligodendrocyte progenitor cell (OPC) death [10], we next sought to investigate the effect of LPS administration on OPCs. The percentage of OPCs was substantially lower in LPS-treated mice 24 h after LPS administration compared to their control counterparts. For Bl6 mice, the decline of OPCs was statistically significant solely in the cerebellum. On the contrary, 129Sv mice exhibited significant decline in OPCs only in the hippocampus. It should be noted, however, that the same downtrend was seen in both brain regions in both strains (Figure 4).

Thereafter, we questioned whether there could be strain-specific differences in microglial activation in response to LPS. Thus, we chose to investigate LPS-induced changes in fractalkine receptor (CX3CR1), cluster of differentiation molecule 11b (CD11b), major histocompatibility complex II (MHC-II), and mannose receptor (CD206).

It has been suggested previously that CX3CR1-CX3CL1 interaction serves as a communication tool between neurons and microglia and could play a role in the microglial activation process [11,12]. To evaluate the LPS-induced alterations in CX3CR1 expression level, we quantified the mean fluorescence intensity (MFI) level of CX3CR1 on microglial cells (Figure 5A–C). LPS induced significant elevation of CX3CR1 expression in the cerebellum of 129Sv mice, whereas no alterations were observed in the hippocampus. In Bl6 mice, LPS did not alter CX3CR1 surface expression.

Furthermore, we found a significant increase of CD11b-reactive microglia in response to LPS stimulation in the hippocampus and cerebellum of 129Sv mice, whereas LPS did not alter the surface expression of CD11b on microglial cells of Bl6 mice. However, compared to 129Sv, the level of CD11b was higher in Bl6 mice, regardless of the nature of treatment. Elevated expression of CD11b reflects a stable and fundamental activation status of microglia in the Bl6 strain.

To study the polarization of the proinflammatory microglial ratio after systemic LPS challenge, we evaluated MHC-II and CD206 expression on microglial cells. Surprisingly, the percentage of MHC-II on microglial cells was significantly decreased in response to LPS stimulation in the cerebellum and hippocampus of Bl6 mice. The percentages of CD206 remained unaffected in both Bl6 and 129Sv strains. Accordingly, the MHC-II/CD206 microglial ratio was significantly lower 24 h after LPS administration in Bl6 mice, whereas no alterations were observed in 129Sv mice (Figure 5G–L).

### 3.4. Impact of LPS on mRNA Levels of Genes Related to the MHC-I Pathway

Since we, surprisingly, observed LPS-induced downregulation of MHC-II in the brain of Bl6 mice, we next sought to investigate mRNA levels of genes related to the MHC-I pathway. We focused on the following genes: β-2-microglobulin (*β2m*), transporter associated with antigen processing (TAP) subunits 1 and 2 (*Tap1* and *Tap2*), the bridging factor tapasin (*Tapbp*), and immunoproteasome subunit *Lmp2*. Gene expression analysis was carried out in six different brain regions: hippocampus, hypothalamus, midbrain, frontal cortex, olfactory bulb, and cerebellum.

In the hippocampus, hypothalamus, and midbrain we observed a significantly higher baseline level of *β2m* mRNA in saline-treated Bl6 control mice compared to 129Sv control mice. Additionally, in the hypothalamus the baseline level of the *tapbp* gene was also higher in Bl6 compared to 129Sv mice.

LPS treatment significantly increased the expression of all MHC-I-related genes (*Tapbp*, *β2m*, *Tap1*, *Tap2*), apart from immunoproteasome gene *Lmp2*, in both strains compared to their respective saline-controls (Figure 6; all *p* < 0.05). However, when comparing Bl6 and 129Sv mice, LPS-induced increase of these MHC-I components was significantly higher in Bl6 mice.

Similarly, LPS increased the expression of *Tapbp*, *β2m*, *Tap1*, *Tap2*, and *Lmp2* mRNA in the hypothalamus, midbrain, and frontal cortex of Bl6 and 129Sv mice (Figure 6; all *p* < 0.05 versus corresponding saline treatment group). However, for all genes, the LPS-induced increase of relative mRNA levels was greater in Bl6 mice than in 129Sv mice.

The most substantial strain-specific differences were observed in the olfactory bulbs. As seen in other brain regions, LPS also increased the expression of *Tapbp*, *β2m*, *Tap1*, *Tap2*, and *Lmp2* mRNA in the olfactory bulb of Bl6 mice (Figure 6; all *p* < 0.05 versus saline). However, in the olfactory bulb of 129Sv mice, the expression of MHC-I related genes was not influenced by LPS challenge.

In the cerebellum, we could not detect any expression of *Lmp2* in Bl6 or 129Sv mice. The mRNA level of *Tapbp* was solely upregulated in Bl6 mice after LPS administration (Figure 6; *p* < 0.05 versus saline), and no changes occurred in 129Sv mice. *β2m*, *Tap1*, and *Tap2* were all significantly upregulated in both strains. However, in the cerebellum only the expression of *β2m* was greater in Bl6 than in 129Sv mice after LPS challenge (Figure 6; *p* < 0.05 Bl6 LPS versus 129Sv LPS).

In addition, since the dipeptidase angiotensin converting enzyme (ACE) has been identified as having a physiological role in the processing of peptides for MHC-I, we thereafter questioned whether the mRNA level of *ACE* was also differently affected by LPS. When comparing control and LPS groups, we found that, while *ACE* gene expression in 129Sv mice was not affected by LPS challenge, in Bl6 mice LPS induced a significant increase in the *ACE* mRNA level in the hippocampus, hypothalamus, midbrain, and cerebellum (Figure 6; all brain regions *p* < 0.05 Bl6 saline versus Bl6 LPS). Although the level of *ACE* in the frontal cortex and olfactory bulbs was not statistically significantly increased by LPS treatment, it showed a tendency towards increase in Bl6 mice.

### 3.5. Impact of LPS on Selected MHC-I-Pathway Protein Levels

In order to provide further validation of strain-specific, LPS-induced alterations seen in mRNA levels, we next performed Western blot analysis. We selected *β2m*, *Tapbp*, and *ACE* genes to further explore the protein levels in the hippocampus and olfactory bulbs of Bl6 and 129Sv mice. Although the LPS-induced effects were less prominent at the protein level than that observed at mRNA level, the main effect was, for the most part, similar. We confirmed that the upregulation of ACE and β2m was greater in Bl6 mice compared to 129Sv in the hippocampus (Figure 7A,B,D). However, no significant upregulation was detected in the Tapbp protein level (Figure 7C) in the hippocampus, which was different from that of the transcript level. Western blot results from olfactory bulb tissue were consistent with qPCR results, confirming increase of Tapbp and β2m in Bl6 and no significant change in 129Sv mice after exposure to LPS (Figure 7G,H). No statistically significant change in the expression of ACE protein was observed in the olfactory bulbs of Bl6 and 129Sv. Although similar to qPCR, it showed a tendency towards increase in Bl6 mice (Figure 7F).

### 3.6. Association between Bodyweight, Temperature Change, and Relative mRNA Expression of MHC-I Pathway Components

To find out whether the upregulation of MHC-I genes had any correlation with bodyweight and temperature change, we then analyzed the LPS groups by Pearson’s analysis.

We found that bodyweight change was positively significantly correlated with *β2m* (*r* = 0.77, *p* = 0.009; Figure 8A), *Tapbp* (*r* = 0.74, *p* = 0.01; Figure 8B), and *Tap1* (*r* = 0.66, *p* = 0.04; Figure 8C) in the olfactory bulbs of Bl6 mice and with *Tap1* (*r* = 0.65, *p* = 0.04; Appendix A) in the cerebellum of Bl6 mice. No significant correlations were established in the olfactory bulbs and cerebellum of 129Sv mice (Figure 8F–J). However, we found that bodyweight change was positively significantly correlated with *Tapbp* (*r* = 0.73, *p* = 0.02; Figure 9G), *Tap1* (*r* = 0.71, *p* = 0.02; Figure 9H), and *Lmp2* (*r* = 0.63, *p* = 0.05; Figure 9J) in the frontal cortex of 129Sv mice, while no significant correlations were established in the frontal cortex of Bl6 mice (Figure 9A–E).

## 4. Discussion

Previous studies have shown that different mouse strains display vast dissimilarities in immune responses, and genetic background is known to influence the outcome in mouse models of human disease. To further map out this variability, we compared differences in endotoxin-mediated inflammatory response in two commonly used mouse strains: C57BL/6NTac (Bl6) and 129S6/SvEvTac (129Sv). To stimulate innate immunity and activate neuroimmune and neuroendocrine systems, we used systemic administration of lipopolysaccharide (LPS; 0.5 mg/kg, i.p.). LPS is a major component of the outer membrane of Gram-negative bacteria and serves as an early warning signal of bacterial infection. In the case of infection, LPS is extracted from bacterial membranes and bound to LPS binding protein (LBP), which transfers LPS through interaction with CD14 to toll-like receptor 4 (TLR4). LPS-TLR4 interaction leads to activation of multiple signaling components, including NF-κB and IRF3, and the production of pro-inflammatory cytokines [13,14,15]. LPS is believed not to penetrate blood brain barrier (BBB), and most effects of peripherally administered LPS are mediated through LPS receptors located outside the BBB [16]. However, LPS could infiltrate the brain through areas lacking the BBB, such as circumventricular organs (CVOs) [17].

### 4.1. LPS-Induced Loss of Body Temperature and Weight and Depression of Locomotor Activity

LPS induced a significant decline in bodyweight in both strains 24 h after administration. However, the LPS-induced loss of bodyweight was more substantial in Bl6 compared to 129Sv mice. Reduced appetite and weight loss have been considered common hallmark physiological responses that occur in many infectious and inflammatory diseases. Previous studies have demonstrated that LPS enhances leptin synthesis and secretion, which is a known appetite suppressant [18,19].

Another important feature of endotoxin-mediated inflammation is the change in body temperature. LPS induced a significant reduction of body temperature in Bl6 mice 24 h after administration. However, this hypothermic response was not seen in 129Sv mice. Thermoregulation has been known to be vital for host defense from invading pathogens and can be disadvantageous if impaired. A recent study demonstrated that inflammation-induced hypothermia and hypometabolism are beneficial for host tolerance and improve survival [20]. Furthermore, we have recently shown that in inflammatory conditions, Bl6 mice exhibit a stronger hypometabolic state than 129Sv mice [6]. Moreover, Chisholm and colleagues demonstrated that reducing metabolic demand through hypothermia increased survival of mice during endotoxemia [21]. This could be one of the reasons 129Sv mice seem to be more vulnerable to infections. Furthermore, we established moderate positive significant correlation between bodyweight and temperature loss in Bl6 mice, whereas no correlation was seen in the 129Sv strain.

In addition to weight loss and body temperature changes, diminished locomotor activity is yet another hallmark response associated with sickness behavior. In the event of infection, the energy demands of the immune system increase, and locomotor retardation helps animals conserve energy, which is redirected into maintenance and survival programs [22]. Both Bl6 and 129Sv strains demonstrated an overall similar pattern of motor activity depression in the 24 h period after LPS-induced inflammation. LPS significantly suppressed the total distance traveled in the whole arena and in the center zone compared to mice in the control group. In the whole arena, a similar LPS-induced suppressive effect was observed in light and dark phases of the 24 h cycle. Whereas in the central area, LPS-induced motor suppression was evident only in the dark phase.

In addition, LPS suppressed the time spent moving in the whole arena as well as in peripheral and central zones in both strains compared to their control counterparts. Although LPS decreased the cumulative time spent moving in both strains, saline-treated Bl6 control mice spent more time moving in the center of the arena, compared to 129Sv control mice, indicating higher anxiety-like baseline behavior of 129Sv mice. The higher anxiety-like trait of 129Sv mice has been thoroughly investigated and well-characterized by several studies [1,2,3].

It should be noted that the motor activity experiment was performed with a separate group of animals to determine whether the changes seen in body temperature and bodyweight are related to changes in motor activity. Since in a 24 h period the motor activity is relatively similar in 129Sv and Bl6 animals, it does not explain the significant differences in body temperature and weight established in these strains. Moreover, we did not detect any significant correlation between 24 h locomotor activity parameters and bodyweight and temperature change.

### 4.2. Microglial Profile and Subpopulations

We chose two brain regions to investigate cell populations and microglial profile—the hippocampus and the cerebellum. The cerebellum is suggested to have an increased sensitivity to circulating inflammatory substances because of its microvascular structure [23] and has an important role within many pathological processes [24,25]. It has been shown recently that microglia in the hippocampus and cerebellum have a higher rate of turnover than those in other brain parts and exist in a more immune-vigilant state [26,27].

The central nervous system (CNS) has been described to be immune-privileged, as the blood brain barrier (BBB) limits the migration of peripheral immune cells into CNS. However, in the case of inflammatory stimulus, the BBB may be compromised, and peripheral immune cells can infiltrate the CNS. As expected, 24 h after the LPS challenge, we observed an increase in infiltrating neutrophils and macrophages as well as microglial cells in the brains of both strains, however the increase in microglial cells was only statistically significant in Bl6 mice. We also investigated the effect of LPS on oligodendrocyte progenitor cells (OPCs), since previous studies have demonstrated that upon activation, microglia mediate OPC death [10]. The percentage of OPCs was substantially lower in LPS-treated mice 24 h after LPS administration compared to their control counterparts. For Bl6 mice, the decline of OPCs was statistically significant in the cerebellum and for 129Sv in the hippocampus, although the same downtrend was seen in both brain regions of both strains. Pang and colleagues demonstrated that LPS-activated microglia are deleterious to OPCs. They showed that cell damage occurs within 24 h after LPS treatment and is mediated by nitric-oxide-dependent oxidative damage. They also demonstrated that this damage can be prevented by inhibition of nitric oxide synthase (NOS) [10]. Moreover, we have previously shown that LPS-treated Bl6 mice have elevated plasma levels of endogenous NOS inhibitors ADMA and SDMA [4].

Next, we investigated the profile of microglial activation in Bl6 and 129Sv mice after exposure to LPS. We observed that under control conditions Bl6 mice had significantly higher expression of CD11b on microglial cells compared to 129Sv mice. However, after LPS stimulation the surface expression of CD11b on microglial cells was increased only in 129Sv mice. Although CD11b is constitutively expressed by microglia, its expression is upregulated under inflammatory conditions. Thus, elevated expression of CD11b reflects the activation status of microglia. This could imply that microglia in Bl6 mice are in a higher immune-alert state under physiological conditions compared to 129Sv mice. Furthermore, we detected a significant increase in microglial surface expression of CX3CR1 in the cerebellum of 129Sv mice in response to LPS stimulation, whereas no alterations were observed in Bl6 mice. CX3CR1 is a chemokine receptor that binds to its ligand fractalkine. Their interaction is associated with crosstalk between neurons and microglia and could play a role in the microglial activation process [11,12]. Recent evidence suggests that CX3CR1 is associated with inflammatory response in the brain of hypertensive animal models, and inhibition of CX3CR1 microglia signaling attenuates hypertension and chronic brain inflammation [28]. This could suggest that LPS may induce hypertension and a higher inflammatory state in 129Sv mice.

Phenotypic markers such as MHC-II and CD206 have been widely used to identify classically activated M1 and alternatively activated M2 (respectively) polarized microglia cells [29,30]. However, it is important to note that M1/M2 classification is highly simplified and does not quite correspond to the variety of microglia phenotypes. Significant percentages of CD206+ microglial cells remained unaffected in both Bl6 and 129Sv strains after LPS stimulation. Interestingly, we observed an immune-suppressive response in the hippocampus and cerebellum of Bl6 mice, which was characterized by the downregulation of the MHC-II+ microglial cell percentage. Moreover, when comparing the MHC-II/CD206+ microglial ratio, we found that the ratio was decreased in response to LPS in Bl6 mice, whereas no alterations were observed in 129Sv mice. There are somewhat conflicting data regarding microglial MHC-II regulation after LPS administration. Though most studies report upregulation of MHC-II+ microglia after exposure to LPS, a recent study described a specific immune-suppressive response in midbrain microglia under inflammatory conditions that was characterized by downregulation of MHC-II microglial expression and upregulation of anti-inflammatory cytokines [31].

These data suggest that microglia of 129Sv mice have increased inflammatory status in response to activation by LPS, whereas microglia of Bl6 mice seem to be in a higher baseline immune-alert state. Furthermore, the downregulation of MHC-II could reflect a protective reaction to inflammatory response aimed to prevent cell damage in Bl6 mice.

### 4.3. Impact of LPS on mRNA and Protein Levels of Genes Related to the MHC-I Pathway

The major histocompatibility (MHC) class I molecules are loaded with peptides derived from exogenous sources (mainly of viral origin) and displayed on the plasma membrane of the cell to communicate with cytotoxic T-cells, thereby regulating their effector or regulatory functions. Besides their classically known function of antigen presentation for T-cell activation, MHC-I molecules have been demonstrated to negatively regulate Toll-like receptor (TLR)-triggered inflammatory responses. Recent evidence suggests that MHC-I reverse signaling is involved in regulation of the defense against bacterial and viral infections [32]. Furthermore, MHC-I molecules have been found to attenuate innate inflammatory responses, which protect mice from sepsis [33]. Interaction between MHC-I-expressing cells and cytotoxic T cells has been demonstrated to suppress the TLR-triggered cytokine production [33].

Since we observed LPS-induced downregulation of MHC-II+ microglial cells in Bl6 mice, we questioned whether the MHC-I pathway would be differently altered after exposure to LPS. We conducted RT-qPCR analysis and investigated relative mRNA levels of the following MHC-I-pathway associated genes: β-2-Microglobulin (*β2m*), transporter associated with antigen processing subunits 1 and 2 (*Tap1* and *Tap2*), the bridging factor tapasin (*Tapbp*), and immunoproteasome subunit *Lmp2* in the hippocampus, hypothalamus, midbrain, frontal cortex, olfactory bulb, and cerebellum.

Examination of MHC-I-pathway related gene expression analysis revealed that LPS administration induces an increase in all components of the MHC-I pathway in the hippocampus, hypothalamus, midbrain, frontal cortex, and cerebellum in both strains. However, Bl6 mice exhibited greater LPS-induced upregulation of these genes compared to 129Sv mice. The most substantial difference was seen in the olfactory bulb, which exhibited LPS-induced upregulation of MHC-I-pathway related genes only in Bl6 mice but remained unaffected by LPS challenge in 129Sv mice. This could mean that the olfactory bulb of Bl6 mice is in a more immunocompetent state than that of 129Sv mice.

In addition, we also investigated the expression of angiotensin converting enzyme (*ACE*). ACE controls blood pressure by catalyzing the conversion of angiotensin I to the active vasoconstrictor angiotensin II, which increases blood pressure by causing blood vessels to constrict. ACE is a major component of the renin angiotensin system (RAS) in the brain. We found that, while *ACE* gene expression in 129Sv mice was not affected by LPS challenge, in Bl6 mice LPS induced a significant increase in *ACE* mRNA levels in the hippocampus, hypothalamus, midbrain, and cerebellum. ACE plays a vital role in the immune responses of myeloid cells and is important for neutrophil immune response to bacterial infection. Previous studies have shown that ACE is required for normal neutrophil antibacterial activity, and upregulation of ACE in neutrophils enhances antibacterial immunity in mice [34]. Thus, it seems that greater upregulation of *ACE* and MHC-I pathway genes is beneficial for host defense and may be advantageous in defense against bacteria in Bl6 mice.

We next performed Western blot analysis to characterize the expression of β2m, Tapbp, and ACE at the protein level in the hippocampus and olfactory bulb of Bl6 and 129Sv mice. β2m is an essential component of the MHC-I complex, being part of the cell surface of the functional MHC-I heterodimer. Tapbp is another critical factor in the MHC-I pathway, required for the assembly of MHC-I heterodimers with peptides in the endoplasmic reticulum (ER). Besides their significance in the MHC-I pathway, β2m and Tapbp were selected for protein analysis since we observed differences in their basal gene expression levels between Bl6 and 129Sv mice. Additionally, we selected ACE for further analysis since LPS induced upregulation of *ACE* specifically in Bl6 but remained unaffected by LPS challenge in 129Sv mice.

LPS stimulation upregulated the protein expression of ACE and β2m in the hippocampus and Tapbp and β2m in olfactory bulbs of Bl6 mice. No statistically significant alterations were detected in either brain region of 129Sv mice. These results give further support to the idea that LPS causes stronger upregulation of MHC-I pathway components in Bl6 mice compared to 129Sv mice.

### 4.4. Association between Bodyweight, Temperature Change, and Relative mRNA Expression of MHC-I Pathway Components

Since we observed a significant upregulation in MHC-I pathway genes, we next sought to investigate the associations between bodyweight and temperature change with MHC-I gene expression. Most interestingly, when aligning the bodyweight and temperature change in the LPS-treated mice with their MHC-I gene expressions, we observed significant positive correlations between bodyweight decline and *β2m*, *Tapbp*, and *Tap1* expressions in the olfactory bulbs of Bl6 mice, whereas in 129Sv mice bodyweight decline was positively correlated with *Tapbp*, *Tap1*, and *Lmp2* expressions in the frontal cortex.

This finding is especially interesting since the rodent frontal cortex is known to have a major role in stress-related behaviors and is involved in determining coping outcomes. It has been demonstrated repeatedly that Bl6 and 129Sv mice display clearly different coping strategies in response to different stressors. Bl6 mice have an active coping strategy in stressful situations, whereas the coping strategy of 129Sv strain is considered passive [4]. Furthermore, we have previously demonstrated that Bl6 mice seem to cope actively with inflammation by inducing a stronger hypometabolic state, whereas the metabolism of 129Sv mice appears to enhance the proinflammatory status [6].

On the other hand, the olfactory bulb is considered to be an exceptional brain structure in terms of adult neurogenesis. Neurons in the olfactory bulb are constantly replaced by new neurons generated in the subventricular zone of lateral ventricles. Mice rely on their sense of olfaction to locate and identify food sources, engage in social interactions, and avoid predators. Moreover, behaviors like learning and memory, social interaction, fear, and anxiety are closely related to their olfactory function. Previous work has demonstrated that the exposure of mice to cat odor induces anxiety responses from Bl6 mice but does not stimulate such responses in the 129Sv strain [35]. This demonstrates different olfactory and anxiety responses in these two mouse strains. Moreover, previous findings have suggested that olfactory bulb microglia may function as tolerogenic antigen-presenting cells (APCs) [36]. The fact that LPS induces significant upregulation of MHC-I genes in olfactory bulbs of Bl6 mice and no changes in olfactory bulbs of 129Sv suggests that the olfactory bulb of Bl6 mice are in a more immunocompetent state than in 129Sv mice.

It should be considered that there may also be differences in the gut microbiota of these strains that we are unaware of. Gut microbiota can influence immune system development and function and can contribute to diverse immunological response [37]. Though, we believe that since these mice have been living in the same conditions for several generations and eating the same food, the specific microbiota that they have acquired is due to their own genetic and immunological peculiarities. A recent study demonstrated that the microbiota regulates microglia function through TLR4, priming these cells to respond to infection [38]. Future research should be considered to investigate differences in microbiota and its potential effects on the immunological response of these strains.

## 5. Conclusions

In the present study, we utilized different methods to examine possible differences in the profile of microglia activation and overall severity of neuroinflammatory status of Bl6 and 129Sv mouse strains in response to LPS stimulation. LPS induced a similar decline in 24 h motor activity in both stains. Both stains also exhibited significant bodyweight loss due to LPS treatment, although it was more severe in Bl6. Furthermore, LPS-induced a hypothermic response in Bl6 mice but not in 129Sv. Differently from Bl6, microglia of 129Sv mice had increased inflammatory status in response to activation by LPS. However, we did see that under baseline conditions, microglia of Bl6 mice seem to be in a higher immune-alert state. Gene and protein expression analysis revealed that LPS administration induced a significantly stronger upregulation of MHC-I-pathway related components in the brain of Bl6 compared to 129Sv mice. Additionally, correlation analysis highlighted the olfactory bulb region of Bl6 mice and the frontal cortex of 129Sv mice as brain regions most affected by LPS in these strains. Based on these results, we hypothesize that the brain of Bl6 mice, particularly the olfactory bulb region, exists in a more immunocompetent state compared to that of 129Sv mice. Several studies have demonstrated that Bl6 and 129Sv mice respond differently to different types of stressful manipulations and have characterized Bl6 mice as an actively coping strain. Findings of our study further support the concept that Bl6 mice are active copers even from an immunological point of view.

## Figures and Tables

**Figure 1 cells-11-01032-f001:**
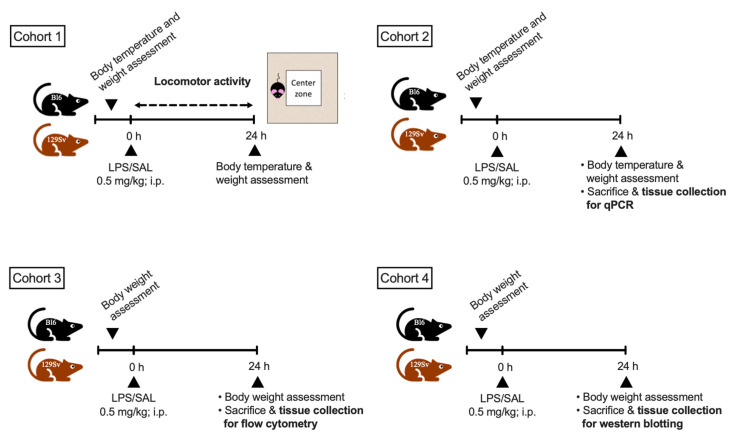
Schematic overview of the experimental cohorts.

**Figure 2 cells-11-01032-f002:**
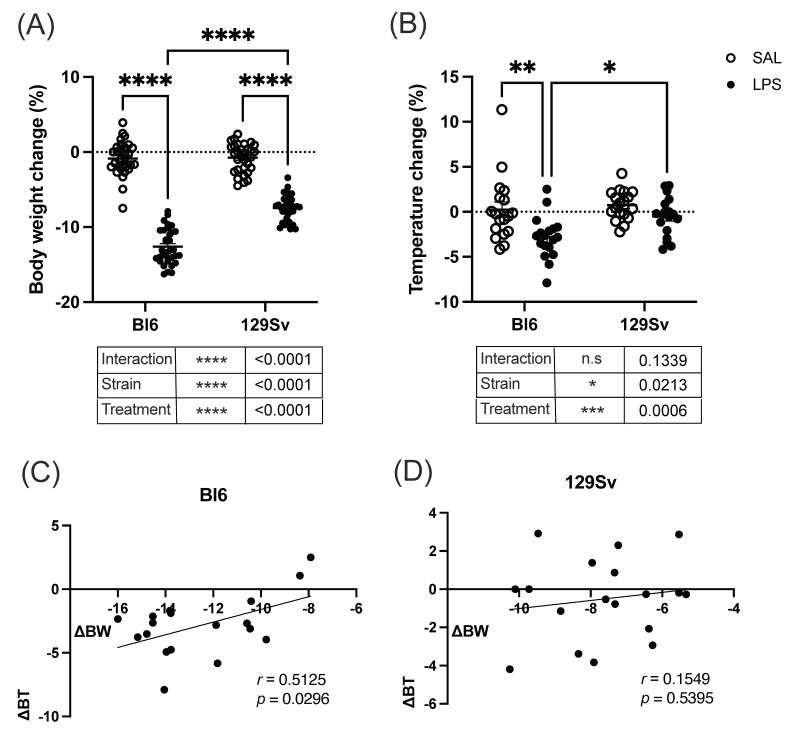
LPS-induced bodyweight and temperature alterations. The bodyweight and temperature of Bl6 and 129Sv mice was measured before and after 24 h of LPS (0.5 mg/kg, i.p) and saline administration. (**A**) Bodyweight change (%). (**B**) Body temperature change (%). Pearson correlation between bodyweight (ΔBW) and body temperature change (ΔBT) in Bl6 (**C**) and 129Sv (**D**) mice. Data are expressed as mean values ± SEM: n.s. > 0.05, * *p* ≤ 0.05, ** *p* ≤ 0.01, *** *p* ≤ 0.001, **** *p* ≤ 0.0001.

**Figure 3 cells-11-01032-f003:**
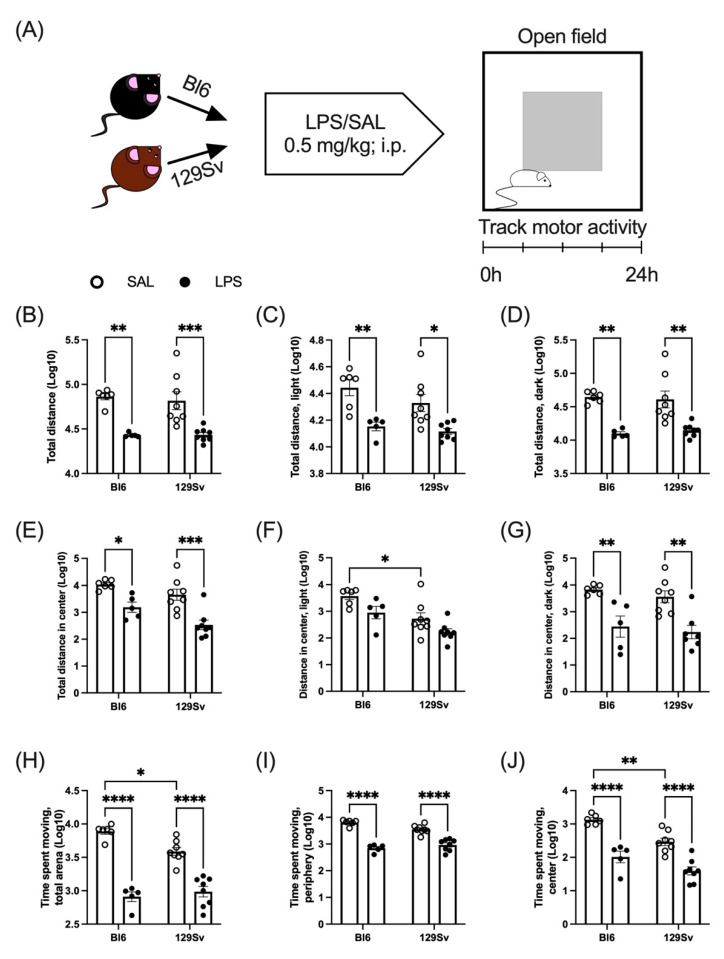
The effect of LPS administration on locomotor activity (Log10 values, data expressed as mean ± SEM). (**A**) Schematic overview of the experiment: Bl6 (*n* = 16) and 129Sv (*n* = 16) mice were injected with LPS (0.5 mg/kg; i.p), and locomotor activity was recorded for 24 h in PhenoTyper cages. Total distance traveled in 24 h (**B**), lights-on (**C**), and lights-off (**D**) period. Total distance traveled in the center of the arena in 24 h cycle (**E**), lights-on (**F**), and lights-off (**G**) periods. Time spent moving in the whole arena (**H**), peripheral (**I**), and central (**J**) zones. Data was analyzed by two-way ANOVA followed by the Bonferroni post hoc test: * *p* ≤ 0.05, ** *p* ≤ 0.01, *** *p* ≤ 0.001; **** *p* ≤ 0.0001.

**Figure 4 cells-11-01032-f004:**
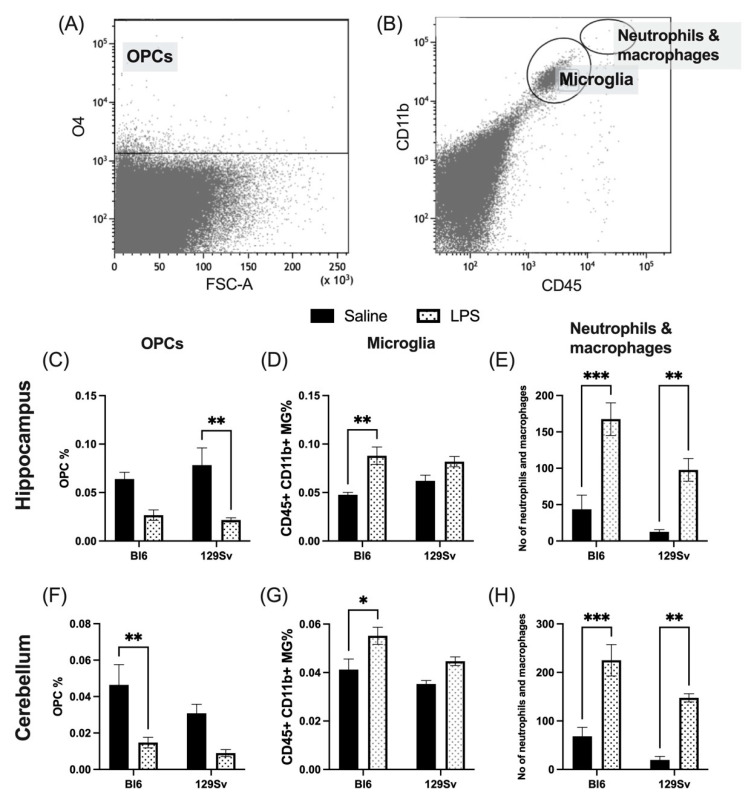
LPS-induced alterations in cell abundance of oligodendrocyte progenitor cells (OPCs), microglia, and neutrophils and macrophages within the brains of Bl6 and 129Sv mice. A representative gating strategy for (**A**) O4 positive OPC population, (**B**) CD11b+CD45^intermediate^ microglia population, and CD11b+CD45^high^ neutrophil and macrophage population. (**C**) Percentage of OPCs, (**D**) microglial cells, and (**E**) number of neutrophils and macrophages in hippocampus. (**F**) Percentage of OPCs, (**G**) microglial cells, and (**H**) number of neutrophils and macrophages in cerebellum. Data are expressed as mean values ± SEM. Data was analyzed by two-way ANOVA followed by the Bonferroni post hoc test: * *p* ≤ 0.05, ** *p* ≤ 0.01, *** *p* ≤ 0.001.

**Figure 5 cells-11-01032-f005:**
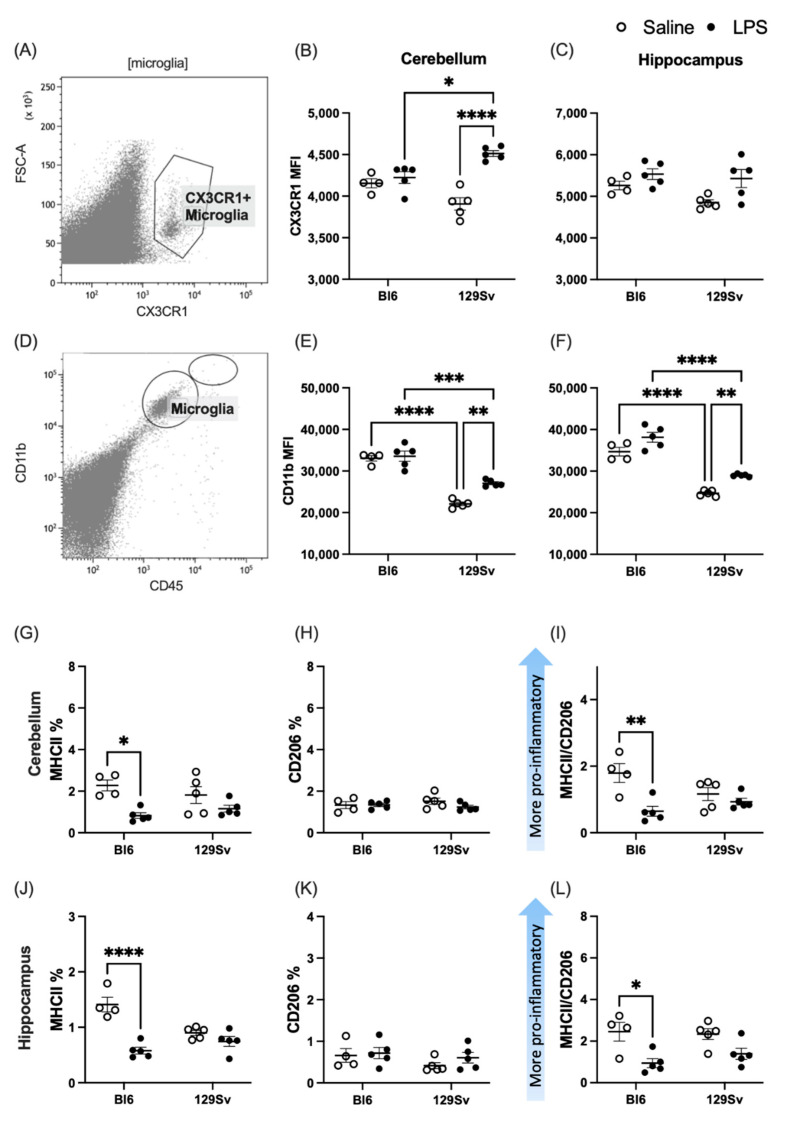
Microglial surface expression of CX3XR1, CD11b, MHC-II, and CD206 molecules. A representative gating strategy for CX3CR1-(**A**) and CD11b-(**D**) positive microglia population. MFI of CX3CR1 in cerebellum (**B**) and hippocampus (**C**). MFI of CD11b in cerebellum (**E**) and hippocampus (**F**). Percentages of (**G**) MHC-II and (**H**) CD206 and (**I**) ratios of MHC-II versus CD206 microglia in cerebellum. Percentages of (**J**) MHC-II and (**K**) CD206 and (**L**) ratios of MHC-II versus CD206 microglia in the hippocampus: * *p* ≤ 0.05, ** *p* ≤ 0.01, *** *p* ≤ 0.001, **** *p* ≤ 0.0001.

**Figure 6 cells-11-01032-f006:**
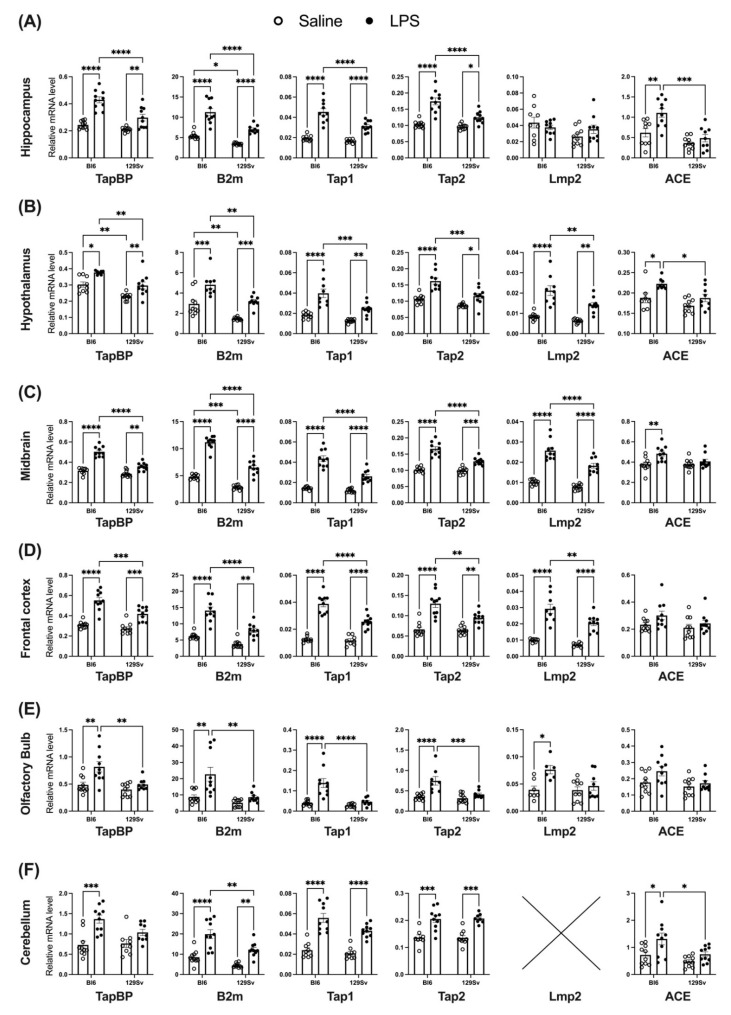
The mRNA expression of MHC-I-related genes in hippocampus (**A**), hypothalamus (**B**), midbrain (**C**), frontal cortex (**D**), olfactory bulb (**E**), and cerebellum (**F**): ×—no expression of *Lmp2* was detected in the cerebellum. Data were analyzed by two-way ANOVA with Bonferroni post hoc test and presented as mean ± SEM: * *p* ≤ 0.05, ** *p* ≤ 0.01, *** *p* ≤ 0.001, **** *p* ≤ 0.0001.

**Figure 7 cells-11-01032-f007:**
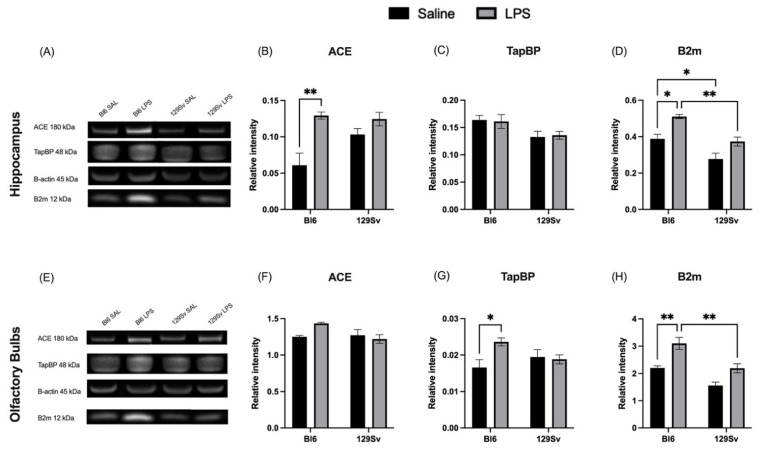
Representative immunoblots (**A**) and protein expression analysis of ACE (**B**), Tapbp (**C**), and β2m (**D**) in hippocampus. Representative immunoblots (**E**) and protein expression analysis of ACE (**F**), Tapbp (**G**), and β2m (**H**) in olfactory bulbs. Data were analyzed by two-way ANOVA with Bonferroni post hoc test and presented as mean ± SEM (*n* = 3–6): * *p* ≤ 0.05, ** *p* ≤ 0.01.

**Figure 8 cells-11-01032-f008:**
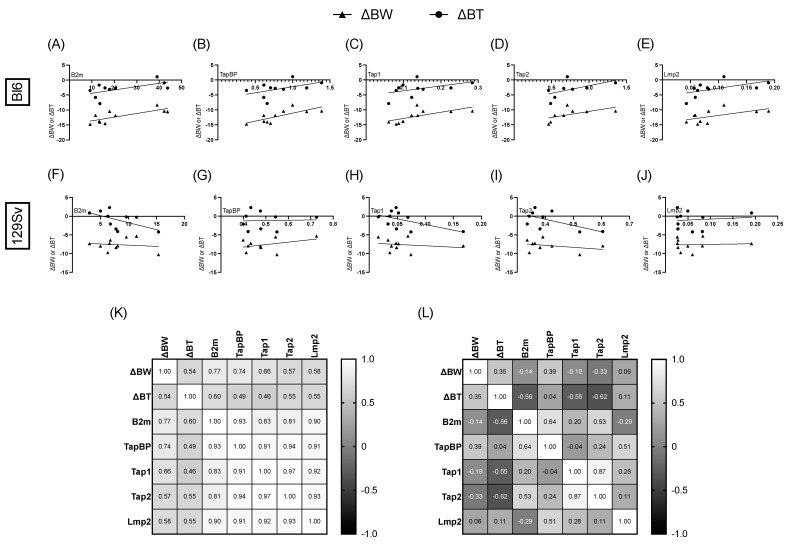
Correlation of MHC-I pathway gene expressions with bodyweight (ΔBW) and temperature (ΔBT) change in olfactory bulb. Pearson correlation of ΔBW and ΔBT with *β2m* (**A**), *Tapbp* (**B**), *Tap1* (**C**), *Tap2* (**D**), and *LmI*(**E**) in Bl6 mice. Pearson correlation of ΔBW and ΔBT with *β2m* (**F**), *Tapbp* (**G**), *Tap1* (**H**), *Tap2* (**I**), and *Lmp2* (**J**) in 129Sv mice. Heatmap of Pearson correlation coefficients between ΔBW, ΔBT, and MHC-I-pathway gene expressions in Bl6 (**K**) and 129Sv (**L**) mice.

**Figure 9 cells-11-01032-f009:**
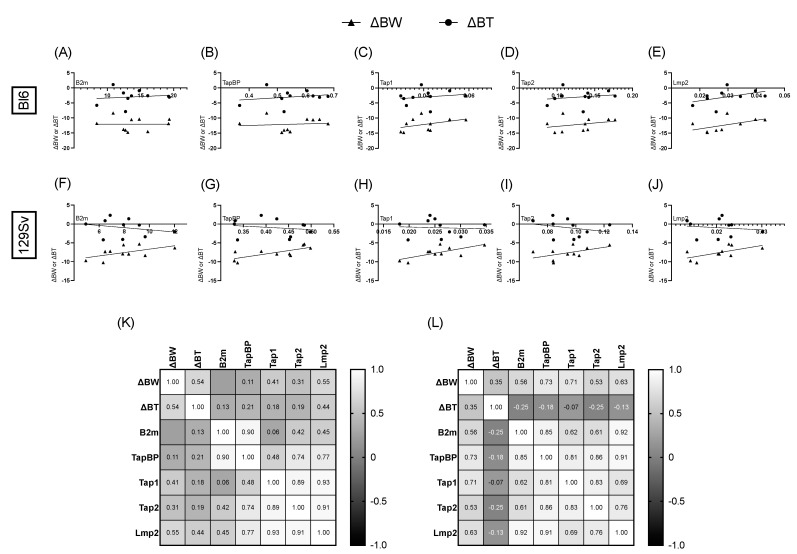
Correlation of MHC-I pathway gene expressions with bodyweight (ΔBW) and temperature (ΔBT) change in frontal cortex. Pearson correlation of ΔBW and ΔBT with *β2m* (**A**), *Tapbp* (**B**), *Tap1* (**C**), *Tap2* (**D**), andI*p2* (**E**) in Bl6 mice. Pearson correlation of ΔBW and ΔBT with *β2m* (**F**), *Tapbp* (**G**), *Tap1* (**H**), *Tap2* (**I**), and *Lmp2* (**J**) in 129Sv mice. Heatmap of Pearson correlation coefficients between ΔBW, ΔBT, and MHC-I-pathway gene expressions in Bl6 (**K**) and 129Sv (**L**) mice.

**Table 1 cells-11-01032-t001:** List of DNA primer sequences used in RT-qPCR analysis.

Gene		Sequence (5′-3′)	Product Length (bp)
*Hprt*	Forward	5′-GCAGTACAGCCCCAAAATGG-3′	85 bp
	Reverse	5′-AACAAAGTCTGGCCTGTATCCAA-3′	
*β2m*	Forward	5′-TGGTCTTTCTGGTGCTTGTC-3′	107 bp
	Reverse	5′-TATGTTCGGCTTCCCATTCTCC-3′	
*Tap1*	Forward	5′-CTTGGATGATGCCACCAGTG-3′	99 bp
	Reverse	5′-AGAAGAACCGTCCGAGAAGC-3′	
*Tap2*	Forward	5′-GCGCCATCTTTTTCATGTGC-3′	143 bp
	Reverse	5′-AAGGTCTTGGCGCAACAAAG-3′	
*Lmp2*	Forward	5′-ATGGGAGGGATGCTAATTCGAC-3′	134 bp
	Reverse	5′-ATGGCATCTGTGGTGAAACG-3′	
*Tapbp*	Forward	5′-CAGCTACCTCCAGTCACTGC-3′	124 bp
	Reverse	5′-GCCCTGAGAAGCCTGCCA-3′	
*ACE*	Forward	5′-TCGCTACAACTTCGACTGGT-3′	163 bp
	Reverse	5′-GAACTGGAACTGCAGCACAA-3′	

**Table 2 cells-11-01032-t002:** List of antibodies used for Western blotting.

Protein	Host	Dilution	Company	Catalog No
Beta 2 microglobulin (β2m)	Rabbit	1:10,000	Abcam	ab75853
Tapasin (TapBP)	Rabbit	1:1000	Abcam	ab196764
Angiotensin Converting Enzyme 1 (ACE)	Rabbit	1:10,000	Abcam	ab254222
B-actin	Mouse	1:10,000	Santa Cruz Biotechnology	sc-47778

## Data Availability

The data presented in this study are available on request from the corresponding author.

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
