# Peer review of "Lipopolysaccharide-Induced Strain-Specific Differences in Neuroinflammation and MHC-I Pathway Regulation in the Brains of Bl6 and 129Sv Mice"

_cells, 2022, doi:10.3390/cells11061032_

Round 1

Reviewer 1 Report

The manuscript by Piirsalu et al. is focused on the analysis of any possible difference in microglia activation and overall severity of neuroinflammation occurring in two widely used mouse strains, C57BL/6NTac (Bl6) and 129S6/SvEvTac (129Sv) upon acute LPS administration. The authors found some similarities and important differences between the two models. Among the most important differences they showed that 24-hour LPS challenge significantly increased the inflammatory status of microglia in 129Sv mice, while under physiological conditions, microglia of Bl6 seemed to be in a higher immune-alert state. All in all, the authors conclude that the brain of Bl6 mice is in a more immunocompetent state compared to 129Sv mice. The overall work is well performed and described. Before publication, however, some unclear aspects need to be addressed before publication. These recommendations are listed below:

1) The authors conclude that "the brain of Bl6 mice is in a more immunocompetent state compared to 129Sv mice". This reviewer is wondering how and if the microbiota of the two type of murine models have any influence on this "diverse" immunological behavior. I strongly deem that the authors should comment/expand this unclear aspect in their study and manuscript

2)Did the authors evaluated the expression of TLR4 (and maybe also TLR2) in brain immune cells?? As the main receptor of LPS, it surprised me to not find any experiment or mention about, at least, TLR4 expression in the two mice models used.

3) The authors used LPS from Sigma and not an Ultra-pure LPS. This is a crucial point. Did the authors evaluate the total absence of any immunostimulant contaminant in the purchased LPS that might be responsible of some of the effects observed in their experiments?? An SDS-PAGE with specific staining(s) and the results of a colorimetric assay that clearly demonstrate the purity of the LPS must be presented.  

Author Response

We appreciate the time and effort dedicated to providing feedback and thank you for your thoughtful comments and efforts towards improving our manuscript. We have incorporated most of the suggestions. Please see below, in bold, for a point-by-point response to the comments and concerns. 

1) The authors conclude that "the brain of Bl6 mice is in a more immunocompetent state compared to 129Sv mice". This reviewer is wondering how and if the microbiota of the two type of murine models have any influence on this "diverse" immunological behavior. I strongly deem that the authors should comment/expand this unclear aspect in their study and manuscript

I completely agree with the reviewer that microbiota can influence immunological behavior. Indeed, there may be differences in the microbiota of these strains that we are unaware of. We have previously shown that Bl6 and 129Sv strains have differences in their metabolomic profile, which may be caused by the differences in their microbiota. These mice have been living for several generations in the same conditions and eating the same food, the specific microbiota that they have acquired, could be due to their own genetic  and immunological peculiarities. It is very possible that this could in some part have an impact on their immunological response. A paragraph focused on this aspect has been added to rows 609-617.

2) Did the authors evaluated the expression of TLR4 (and maybe also TLR2) in brain immune cells?? As the main receptor of LPS, it surprised me to not find any experiment or mention about, at least, TLR4 expression in the two mice models used.

Very important remark! Since we administered LPS intraperitoneally, we did not examine the expression of TLR4 in the brain. LPS is believed not to penetrate BBB and most effects of peripherally administered LPS are mediated through LPS receptors located outside the BBB (Banks and Robinson, 2010). Thus, LPS acts through indirect mechanisms, such as release of substances from the periphery that in turn can cross the BBB. Furthermore, we have previously done pilot studies to choose minimal LPS concentration that generates mild inflammation without disrupting the BBB. For the disruption of BBB, high doses of LPS are required (Banks et al., 2015)

Banks, W. A., and Robinson, S. M. (2010). Minimal Penetration of Lipopolysaccharide Across the Murine Blood-brain Barrier. Brain Behav Immun 24, 102–109. doi:10.1016/j.bbi.2009.09.001.

Banks, W. A., Gray, A. M., Erickson, M. A., Salameh, T. S., Damodarasamy, M., Sheibani, N., et al. (2015). Lipopolysaccharide-induced blood-brain barrier disruption: roles of cyclooxygenase, oxidative stress, neuroinflammation, and elements of the neurovascular unit. Journal of Neuroinflammation 12, 223. doi:10.1186/s12974-015-0434-1.

3) The authors used LPS from Sigma and not an Ultra-pure LPS. This is a crucial point. Did the authors evaluate the total absence of any immunostimulant contaminant in the purchased LPS that might be responsible of some of the effects observed in their experiments?? An SDS-PAGE with specific staining(s) and the results of a colorimetric assay that clearly demonstrate the purity of the LPS must be presented. 

Thank you for pointing this out! The LPS used in this article is purified by gel-filtration chromatography and contains <1% protein according to the manufacturer. We appreciate the suggestion and agree that it would be useful to demonstrate the purity of the LPS. However, such an analysis is beyond the scope of our paper, which aims simply to show the differences in immune activation. We believe that if any trace amount of immunostimulant contaminant is present, it influences both strains equally. 

Reviewer 2 Report

The authors performed a well-designed, -structured and -illustrated study using state-of-the-art methods. They described strain-specific differences in the response to LPS treatment (body temperature, weight, locomotor activity, microglia proliferation/activation, MHC-I pathway). I just have few comments:

  1. Why did the authors show percentages of OPCs and microglia but absolute numbers of neutrophils&macrophages? Is there really a decrease in OPCs in 129Sv mice (Fig. 4C) or is the decreased percentage just the result of increased numbers of other cells such as microglia (Fig. 4D) and neutrophils&macrophages (Fig. 4E)?
  2.  The authors should avoid the terms "M1 microglia" and "M2 microglia". This classification is too simplified and two-dimensional (please see Ransohoff: A polarizing question: do M1 and M2 microglia exist? Nat Neurosci 2016;19:987-91).

Author Response

We appreciate the time and effort dedicated to providing feedback and thank you for your thoughtful comments and efforts towards improving our manuscript. We have incorporated most of the suggestions. Please see below, in bold, for a point-by-point response to the comments and concerns. 

  1. Why did the authors show percentages of OPCs and microglia but absolute numbers of neutrophils&macrophages? Is there really a decrease in OPCs in 129Sv mice (Fig. 4C) or is the decreased percentage just the result of increased numbers of other cells such as microglia (Fig. 4D) and neutrophils&macrophages (Fig. 4E)?

    Thank you for this question! We decided to show absolute numbers of neutrophils & macrophages because there was  such a small percentage of these cells, that the significance did not come out as clearly as when reporting absolute numbers. As for the OPCs, the decrease in OPCs has been reported by several other studies as well.  It has been reported that reduced OPC numbers are due to microglia activation and LPS-activated microglia decrease OPC survival (Miller et al., 2007).

    Miller, B. A., Crum, J. M., Tovar, C. A., Ferguson, A. R., Bresnahan, J. C., and Beattie, M. S. (2007). Developmental stage of oligodendrocytes determines their response to activated microglia in vitro. J Neuroinflammation 4, 28. doi:10.1186/1742-2094-4-28.

  2. The authors should avoid the terms "M1 microglia" and "M2 microglia". This classification is too simplified and two-dimensional (please see Ransohoff: A polarizing question: do M1 and M2 microglia exist? Nat Neurosci 2016;19:987-91).

    I agree that these terms are very simplified. M1/M2 terms have been removed from results and a brief explanation added to discussion section row 494.

Reviewer 3 Report

The manuscript entitled ‘Lipopolysaccharide-induced strain-specific differences in neuroinflammation and MHC-I pathway regulation in the brains 3 of Bl6 and 129Sv mice’ is very extensive, thoroughly prepared and is based on many research methods. However, I have a few questions/comments. Please see below.

1.    l. 63-73: In my opinion, this fragment is redundant in the Introduction section.
2.    What is the purpose of this study? In my opinion, the purpose of this study is not emphasized enough in the Introduction section.  
3.    After reading the Introduction section, I am not sure why these two strains (Bl6 and 129Sv) and such research parameters were chosen?
4.    The Discussion section, divided into sub-chapters, is not very clear, not very synthetic, not summarizing enough. At times, the Discussion focuses too much on the own results and there is no confrontation with the data from the literature.

Author Response

We appreciate the time and effort dedicated to providing feedback and thank you for your thoughtful comments and efforts towards improving our manuscript. We have incorporated most of the suggestions. Please see below, in bold, for a point-by-point response to the comments and concerns. 

  1.    l. 63-73: In my opinion, this fragment is redundant in the Introduction section.

    Thank you for this suggestion. We have removed this section from the introduction.

  2. What is the purpose of this study? In my opinion, the purpose of this study is not emphasized enough in the Introduction section.  
    Answer: We agree with the reviewer’s assessment. We have, accordingly, modified the introduction to emphasize this point. A section emphasizing the importance of this study has been incorporated to rows 39-47 After reading the Introduction section, I am not sure why these two strains (Bl6 and 129Sv) and such research parameters were chosen?

    Bl6 and 129 are one of the most widely used inbred mouse strains in many research fields including immunological studies and obtaining new knowledge regarding their differences is fundamental for appropriate model selection and effective data interpretation. A brief explanation has been added to rows 70-73. 

  3. The Discussion section, divided into sub-chapters, is not very clear, not very synthetic, not summarizing enough. At times, the Discussion focuses too much on the own results and there is no confrontation with the data from the literature.

    Thank you for pointing this out. The rationale behind sub-chapters was to make it easier for readers to navigate through large quantities of information. However, to make it clearer, we have merged some chapters together (Body Temperature, Weight and locomotor activity; mRNA and protein expression of MHC-I). Discussion has been further expanded (rows 451-453, 480-485, 512-517 and 609-617).

Round 2

Reviewer 1 Report

I appreciate the work of authors in trying to solve unclear points raised up by this reviewer. I strongly recommend the publication of their work. However, I also strongly suggest that authors consider the possibility that LPS might enter (totally or in fragments) into the brain through areas lacking the BBB, such as CVOs, as it has been demonstrated by Leza et al. 2017 in rat models. Maybe they should comment this in their manuscript to help non-experts understanding why they did not mention about any TLR4 expression in cells forming the blood-brain interface

Author Response

We agree with the reviewer’s assessment. We have modified the discussion section and incorporated this to rows 416-419.